# Prevalence of intestinal parasite contamination in raw vegetables and school soil samples in Koh Yao, Phang Nga Province, southern Thailand

Udomsak Narkkul[1,2], Prasit Na-ek[1,2], Aulia Rahmi Pawestri[3], Chuchard Punsawad[1,2]*

**1** Department of Medical Science, School of Medicine, Walailak University, Nakhon Si Thammarat, Thailand, **2** Center of Excellence in Tropical Pathobiology, Walailak University, Nakhon Si Thammarat, Thailand, **3** Department of Parasitology, Faculty of Medicine, Universitas Brawijaya, Malang, Indonesia

* chuchard.pu@wu.ac.th

## Abstract

Parasitic infections remain a significant public health issue in Thailand. Exposure to intestinal parasites occurs through consumption of improperly washed raw vegetables and handling of contaminated soil without proper hygiene. This study assessed the prevalence of intestinal parasites in raw vegetables and school soil, along with the associated factors, in Koh Yao, Phang Nga Province, Thailand. Vegetables (21 types) were collected from local markets and stores in Koh Yao Noi, Koh Yao Yai, and Phru Nai. Moreover, soil samples were collected from playgrounds, football fields, sidewalks, schoolyards, and areas around cafeterias in 13 public primary schools. Approximately 2.3% of vegetable samples (3 out of 131) were contaminated with parasites (one peppermint and two Chinese morning glory samples). The most commonly detected parasites were *Ascaris* spp. eggs (1.5%) and hookworms (0.8%). Koh Yao Noi had the highest contamination rate of 7.1%, whereas no contamination was observed in samples from Koh Yao Yai and Phru Nai. The contamination rates in vegetables obtained from the three subdistricts showed significant differences ($p < 0.05$). All contaminated vegetables were grown locally. In soil samples, parasites were detected in 11 of the 13 schools (84.6%). Of the 141 soil samples, 42 (29.8%) were contaminated, and *Toxocara* spp. were found in all contaminated samples. Phru Nai had the highest soil contamination rate (50%), followed by Koh Yao Yai (20.5%), and Koh Yao Noi (11.4%). There were significant differences in the distribution of parasite contamination across the three subdistricts ($p < 0.001$) and seasons ($p < 0.001$). Football fields were the most contaminated areas, with 36% of samples testing positive for parasites. This study highlights the risk of parasitic transmission through contaminated vegetables and soil. Hence, this emphasizes the need for health authorities to educate local communities on proper hygiene practices, including hand washing and thorough vegetable cleaning, to prevent parasitic infections. Implementing preventive measures in schools and communities is crucial for limiting parasitic disease spread.

**Data availability statement:** The data used to support the findings of this study are all included in this article.

**Funding:** This study was supported by the Thailand Science Research and Innovation Fund (Contract No. FRB660041/0227), awarded to UN, PN, CP. The funding body was not involved in the study design, data collection, analysis, interpretation, or writing of the manuscript.

**Competing interests:** The authors declare no competing interests.

## Introduction

Soil-transmitted helminths (STHs) are a group of intestinal parasites that infect approximately 1.5 billion people globally [1], with the highest prevalence in tropical and subtropical regions [2]. The World Health Organization (WHO) identifies these parasites as roundworms (*Ascaris lumbricoides*), whipworms (*Trichuris trichiura*), and hookworms (*Necator americanus* and *Ancylostoma duodenale*) [3,4]. Furthermore, zoonotic parasites, such as *Toxocara* spp. and animal hookworms [5], present major global public health challenges. These parasites are transmitted through the fecal-oral route or via skin penetration, thereby affecting populations with inadequate hygiene and sanitation, particularly children. Infections are either asymptomatic or lead to abdominal pain, diarrhea, malnutrition, anemia, and stunted growth [6]. Several factors, such as family size, source of drinking water, open-field defecation, hand-washing habits, playing with soil, consumption of raw vegetables, and fingernail trimming, were strongly associated with parasitic infections [7]. Foodborne parasitic infections have also been linked to the consumption of contaminated fresh vegetables [8]. Vegetables are generally regarded as effective vehicles for transmitting parasites, particularly when consumed raw or unpeeled [9]. These vegetables, which are often inadequately disinfected, contribute to the spread of parasitic diseases [2]. Additionally, environmental factors, including climate, geography, temperature, soil type, and rainfall play critical roles in the transmission of intestinal parasites [1].

Since 1957, intestinal parasite infections have become a significant public health concern in Thailand. The Ministry of Public Health began a systematic national survey of these infections, conducting assessments every five years. Initially, the prevalence of parasitic intestinal infections was as high as 60%; however, it decreased over time. By 2009, the national prevalence decreased to 18.1%. Regional data revealed that the northeast had the highest prevalence of helminthiasis (26.0%), followed by the south (19.8%) and the north (17.7%). Moreover, a survey conducted in 2014 reported a further decrease of 8.2% [10,11]. In southern Thailand, soil-transmitted parasitic infections, particularly hookworm infections, are the most common parasitic diseases in all age groups [10,12,13]. These infections are transmitted to humans through skin contact or ingestion of parasitic larvae [1].

In southern Thailand, consuming raw fresh vegetables is a widespread practice, reflecting a broader healthy eating trend aimed at preventing major diseases such as cardiovascular conditions and certain cancers [14]. However, raw vegetables are also vectors for intestinal pathogens and, therefore, pose a risk of transmitting human pathogens [15]. Parasitic contamination of fresh vegetables has been reported in various countries, including Arba Minch in Ethiopia [16], Khartoum State in Sudan [17], Benha in Egypt [2], Accra in Ghana [13], Mazandaran in Iran [18], Poland [19], Metro Manila in the Philippines [20], and Thailand [21], particularly southern Thailand [1]. Common helminthic contaminants in these studies include *A. lumbricoides*, hookworms, *T. trichiura*, and *Strongyloides stercoralis* [2,13,16,17]. In the Nakhon Si Thammarat Province, southern Thailand, a survey revealed a 35.1% rate of parasitic contamination in fresh vegetables. The most frequently detected parasite was the hookworm (42.9%), followed by threadworms (*S. stercoralis*; 10.6%), whipworms (*T. trichiura*; 2.6%), roundworms (*A. lumbricoides*; 2.6%), and pet parasites (*Toxocara* spp.; 2.6%) [1]. This contamination is often attributed to irrigation and postharvest handling water. Soil and water are key reservoirs of parasitic cysts and eggs that contribute significantly to the spread of parasitic diseases. Furthermore, the feces of infected animals, such as dogs and cats, introduce these parasites into the environment. Soil contamination by pathogenic parasites is a major public health concern, especially in tropical developing countries, such as Thailand, where the overall prevalence of contamination is approximately 7.8% [22].

Soil is essential for the life cycle of STHs, and their eggs require a warm and moist soil environment to become infective. Hookworm species require up to 14 days to become viable and infectious; *A. lumbricoides* eggs require 8–37 days, *T. trichiura* eggs require 20–100 days [23], and *Toxocara* spp. require 2–4 weeks [24]. STH eggs are excreted in the feces of infected individuals and animals, thereby contaminating the environment of endemic regions. These eggs have been discovered in the soil of rural households in Poland, southern Thailand, the Philippines, Kenya, and primary schools in northern Vietnam [5,25–28], whereas *Toxocara* transmission occurs when infected dogs or cats shed eggs in their feces in public parks or playgrounds [24].

The prevalence of intestinal parasitic infections may be linked to insufficient knowledge of hygiene, poor environmental sanitation, and a lack of awareness regarding the health risks associated with these infections. Islands play a critical role in the dynamics of parasite infection, as their geographic isolation limits the introduction of new parasites and promotes the development of unique endemic strains. Local environmental conditions, human activities, and public health infrastructure significantly influence the prevalence and transmission of these infections within vulnerable populations. Koh Yao, an island located in southern Thailand, is characterized by a warm and humid climate and a dominant agricultural, fishing, and tourism sector. The region faces a growing challenge with a high incidence of intestinal parasites such as hookworm, *S. stercoralis*, *T. trichiura*, *B. hominis*, and *G. lamblia*, with previous reports focusing primarily on human prevalence [12,29,30]. Although a few studies have reported parasitic contamination in vegetables grown and marketed in some southern regions [1], there have been no documented reports of parasitic contamination of vegetables or soil in the Koh Yao area. Consequently, this study aimed to assess the prevalence of intestinal parasite contamination in raw vegetables and school soils, and identify the factors contributing to contamination in Koh Yao, Phang Nga Province, Thailand. The results of this study will guide the development of action plans and community-based initiatives to prevent and control intestinal parasitic diseases in this region. Moreover, these findings will serve as a foundation for future models to manage similar issues in other areas.

## Materials and methods

### Study sites

This cross-sectional study was conducted between November 2022 and May 2023 in the Koh Yao district of Phang Nga Province, southern Thailand, which is approximately 800 km from Bangkok, the capital city. Koh Yao is an island situated in the Andaman Sea within Phang Nga Province. The district encompasses two main islands (Koh Yao Noi and Koh Yao Yai), covers a total area of approximately 137.6 km², and is surrounded by several smaller islands. As of 2020, the total population of Koh Yao was 14,499, resulting in a population density of 112.86 people/km² (292.3 people per square mile). Koh Yao District (Fig 1) is divided into three subdistricts and 18 villages: Koh Yao Noi (seven villages), Koh Yao Yai (four villages), and Phru Nai (seven villages). Koh Yao is a remote area where transportation is primarily by boat, and access to healthcare is limited. The majority of the population is engaged in agriculture and tourism. Previous studies have reported a prevalence of intestinal parasites in humans in this region, ranging from 9.3% to 18.42% [12,29,30], which may be linked to the consumption of unwashed vegetables. Fresh vegetables were purchased from all markets located in the three subdistricts. These vegetables were sourced from various farms and agricultural areas, both on the island and in nearby provinces. Soil samples were collected from all schools across the three subdistricts.

### Sample collection

A total of 133 fresh vegetable samples, representing 21 different types of vegetables that are commonly consumed raw, were randomly purchased from vendors in Koh Yao, Phang,

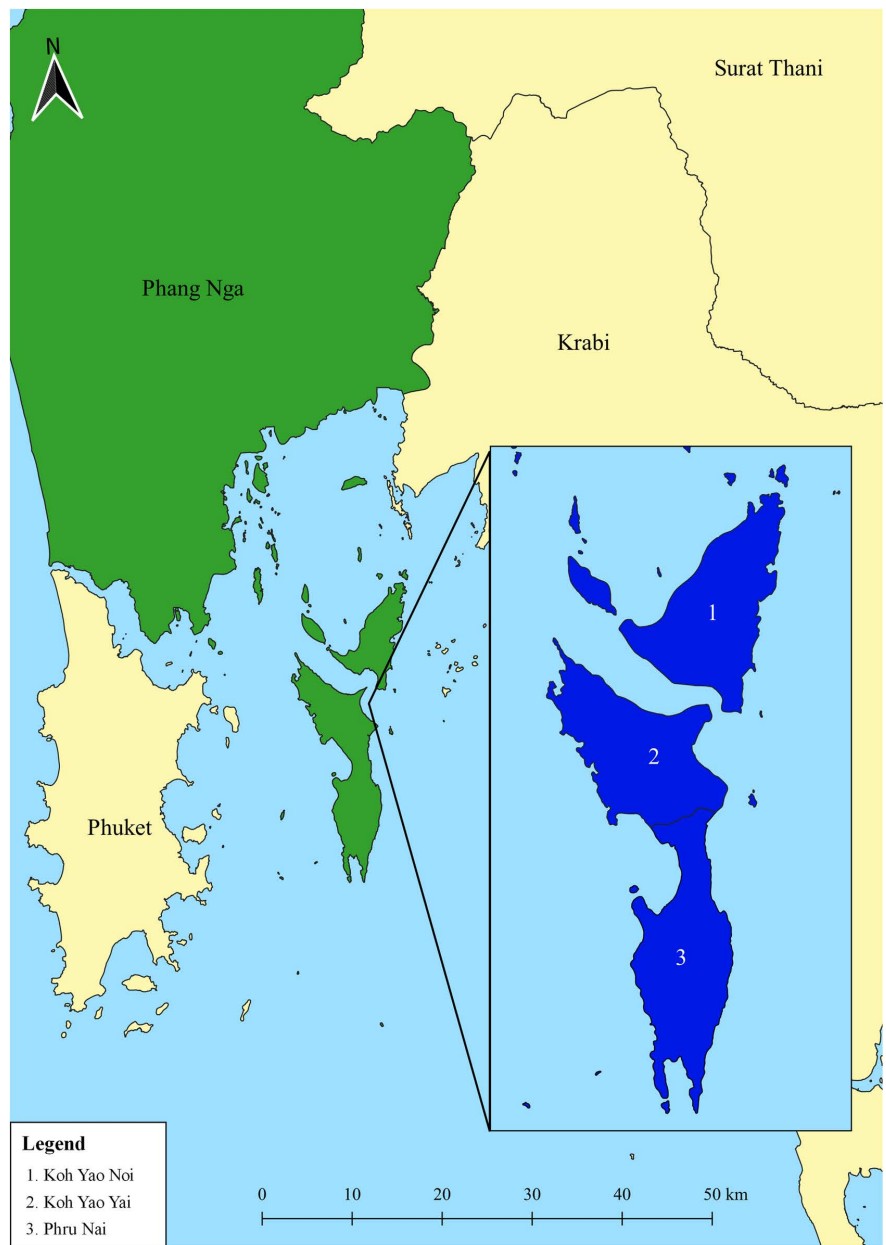

**Fig 1. Study area including the Koh Yao Noi [ 1], Koh Yao Yai [2], and Phru Nai [3] subdistricts of Koh Yao district, Phang Nga Province, southern Thailand.** Quantum GIS version 3.16 (ESRI base maps) was used to generate the map (https://qgis.org/en/site/; accessed on 31 July 2024).

Nga Province. Direct observation data were collected on factors related to parasite contamination in vegetables, including the following: subdistrict, type of fresh vegetables, place of purchase, source of vegetables, presence of animals in the vicinity, characteristics of the area, and hygiene practices (adequately clean or dirty). Hygiene practices assessed the cleanliness of the environment, absence of animals or insects, and characteristics of the soil around the stalls. The vegetable types were classified based on the parts of the plant that are consumed, as shown in Table 1. Photographs of vegetable samples are shown in Fig 2. The samples were collected in clean, labeled plastic bags to avoid contamination and immediately transported

**Table 1. Classification of vegetables based on the parts of the plant that are consumed.**

| Category | Vegetable |
| --- | --- |
| Stems | Leek (*Allium porrum*), Bean sprout (*Pisum sativum*) |
| Leaves | Thai basil (*Ocimum basilicum*), Gotu kola (*Centella asiatica*), Bok choy (*Brassica rapa subsp. chinensis*), Holy basil (*Ocimum tenuiflorum*), Cabbage (*Brassica oleracea*), Celery (*Apium graveolens*), Baby cabbage (*Brassica oleracea*), Chinese cabbage (*Brassica rapa subsp. pekinensis*), Lettuce (*Lactuca sativa*), Coriander (*Coriandrum sativum*), Culantro (*Eryngium foetidum*), Chinese morning glory (*Ipomoea aquatica*), Peppermint (*Mentha × piperita*) |
| Fruits | Cucumber (*Cucumis sativus*), Yardlong bean (*Vigna unguiculata subsp. sesquipedalis*), Winged beans (*Psophocarpus tetragonolobus*), Bitter gourd (*Momordica charantia*), Okra (*Abelmoschus esculentus*), and Thai eggplant (*Solanum virginianum L.*) |

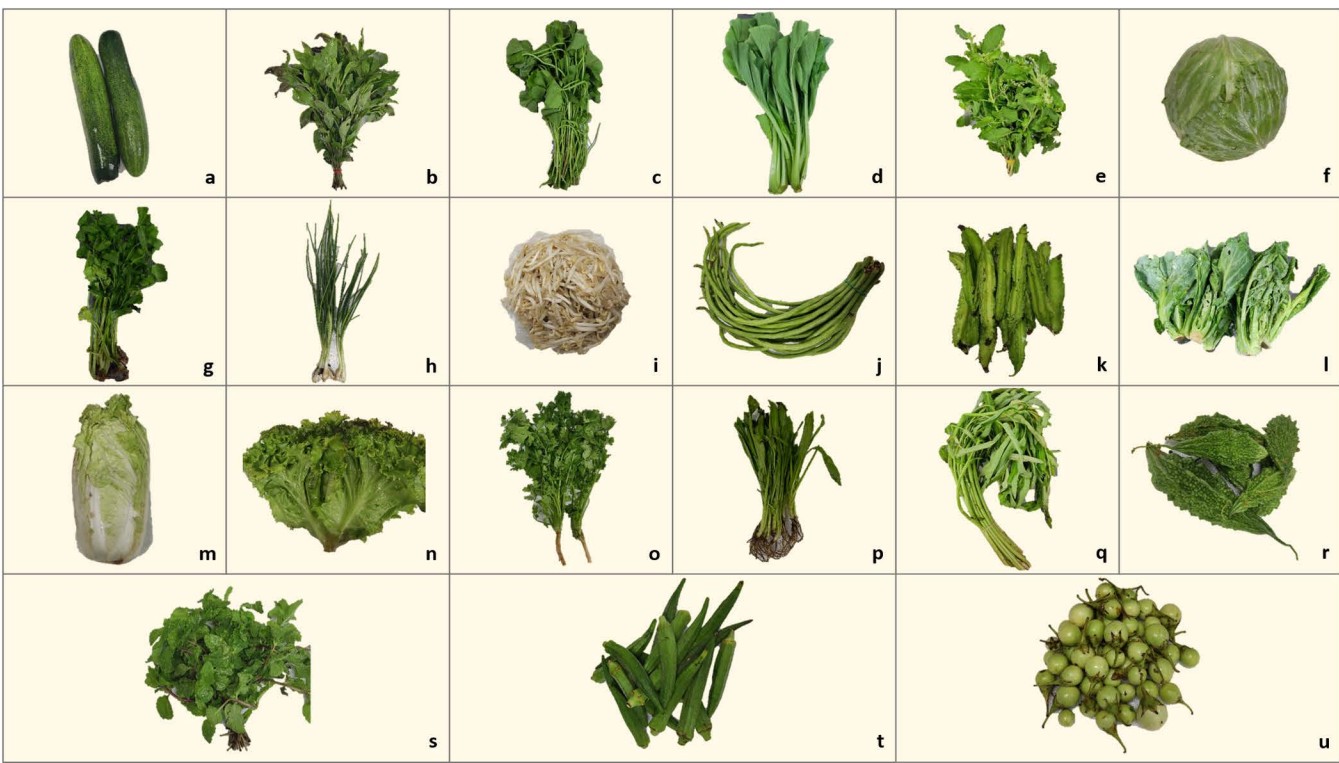

**Fig 2. Photographs of collected vegetable samples.** (a) Cucumber (*Cucumis sativus*), (b) Thai basil (*Ocimum basilicum*), (c) gotu kola (*Centella asiatica*), (d) bok choy (*Brassica rapa subsp. chinensis*), (e) holy basil (*Ocimum tenuiflorum*), (f) cabbage (*Brassica oleracea*), (g) celery (*Apium graveolens*), (h) leek (*Allium porrum*), (i) bean sprout (*Pisum sativum*), (j) yardlong bean (*Vigna unguiculata subsp. sesquipedalis*), (k) winged beans (*Psophocarpus tetragonolobus*), (l) baby cabbage (*Brassica oleracea*), (m) Chinese cabbage (*Brassica rapa subsp. pekinensis*), (n) lettuce (*Lactuca sativa*), (o) coriander (*Coriandrum sativum*), (p) culantro (*Eryngium foetidum*), (q) Chinese morning glory (*Ipomoea aquatica*), (r) bitter gourd (*Momordica charantia*), (s) peppermint (*Mentha × piperita*), (t) okra (*Abelmoschus esculentus*), and (u) Thai eggplant (*Solanum virginianum L.*).

to the parasitology laboratory at the School of Medicine, Walailak University, for detailed parasitic examination. The conditions of the purchasing areas (cleanliness, animals, etc.) were recorded.

A total of 141 soil samples were collected during two distinct seasons to assess variations in contamination levels; one set was collected in the dry season and the other in the rainy season from public schools in the Koh Yao district between November 2022 and May 2023. Direct observation data were collected on factors related to parasite contamination in school soil samples, including the subdistrict, place of collection, presence and type of animals in the vicinity, and soil characteristics. At each school, ten soil samples were collected from five distinct locations where children frequently played or engaged in activities: playgrounds, football fields, sidewalks, schoolyards, and areas around cafeterias (Fig 3). At each location, two samples were collected from separate sites at least 20 m apart, with each sample consisting of 100 g of soil collected from a depth of 0–15 cm [31]. The samples were placed in individually labeled sterile plastic bags and transported to the Parasitological Laboratory at Walailak University. The samples were stored at 4 °C until further analyses.

## Detection of parasites in fresh vegetables

Individual vegetable samples (200 g) were cut into smaller pieces before being washed with 1000 mL of a 0.9% sodium chloride solution, shaken for 15 minutes to dissolve the parasites, and allowed to settle overnight. The supernatant was decanted and the sediment was centrifuged at 2000 × g for 15 min. After removing the supernatant, the sediment was examined under a light microscope using 100 × and 400 × magnification objectives to detect the eggs and

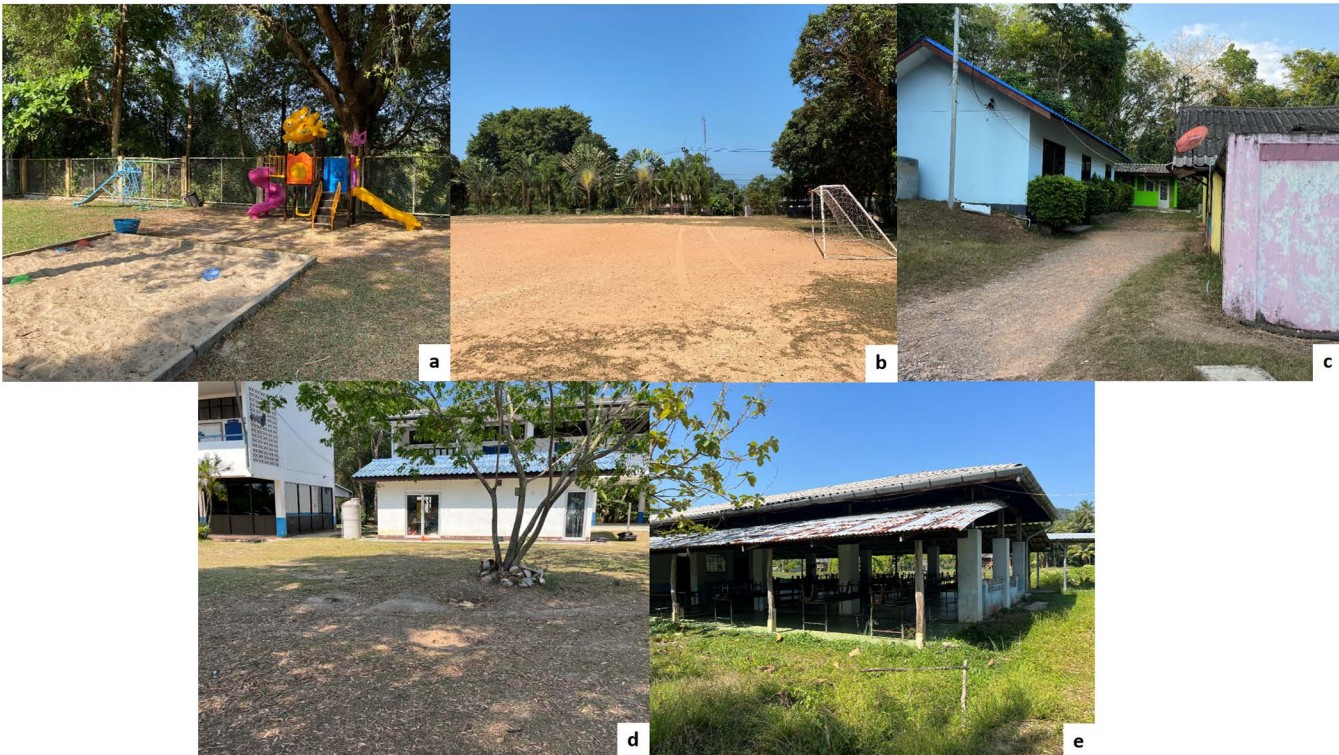

**Fig 3. Photographs of school soil sample collection sites.** (a) Playgrounds, (b) football fields, (c) sidewalks, (d) schoolyards, and (e) areas around cafeterias.

larvae of parasites. The parasites were morphologically identified according to the Centers for Disease Control and Prevention diagnostic reference [32]. To confirm the presence of parasites, each vegetable sample was examined by two trained senior medical laboratory technologists. Three slides per sample were prepared by two independent investigators to enhance parasite detection [1].

### Detection of parasites in soil samples

Soil samples were examined for intestinal parasitic eggs using a modified flotation method with sucrose solution, as previously described [31,33,34]. Briefly, the soil samples were dried overnight at an ambient temperature (26–28 °C) and filtered through a 150-μm mesh sieve. Two grams of powdered soil was mixed with 8 mL of 0.05% Tween-80 solution in a 15-mL tube and centrifuged at 2000 rpm for 10 min. After discarding the supernatant, the sediment was suspended in a sucrose solution (specific gravity of 1.200) to approximately 1 cm from the top of the tube, vortexed, and centrifuged again. The tube was then filled to the top with sucrose solution and centrifuged at 1800 rpm for 5 min. A cover glass was placed on top of each tube and the tubes were examined under a light microscope at 400 × magnification by an experienced medical technologist for intestinal parasitic eggs.

### Statistical analysis

All statistical analyses were conducted using IBM SPSS Statistics for Windows, version 23.0. Qualitative variables were presented as frequencies and percentages. The rate of intestinal parasitic contamination was assessed by determining the percentage with a 95% confidence interval (95% CI). A chi-square test was used to compare the rates of intestinal parasitic contamination in the types of vegetables, subdistricts, sample collection sites, and seasons. Statistical significance was set at $p < 0.05$.

### Ethics approval and consent to participate

This study was reviewed and approved by the Human Research Ethics Committee of Walailak University, Thailand (Approval Number: WUEC-22-329-01), prior to sample collection. Informed consent was not required for the collection of raw vegetable and school soil samples as they were obtained from publicly accessible locations. Permission was obtained from the school authorities for soil sampling on the school premises and from local vendors before collecting vegetable samples.

## Results

A total of 133 fresh vegetable samples were examined for parasitic contamination. Of these, 52 samples (39.1%) came from local stores in the Phru Nai subdistrict, 42 samples (31.6%) from the Koh Yao Noi subdistrict, and 39 samples (29.3%) from the Koh Yao Yai subdistrict (Fig 4). The most frequently tested vegetables included Thai eggplant (10.5%), cabbage (9.8%), leek (9.0%), and Chinese cabbage (7.5%). In terms of the sources of fresh vegetable samples, the majority were community stores (107 samples, 80.4%) and markets (26 samples, 19.6%). Most vegetables were imported from other areas (76 samples, 57.1%), whereas 57 (42.9%) were grown locally. Among the imported vegetables, the majority came from the Krabi province (53.9%), followed by Phuket (26.3%), Nakhon Si Thammarat (14.5%), and Surat Thani (5.3%). In terms of sales areas, most areas did not have pets (91.7%), and where pets were found, only cats were present (100%). The sales areas had cement floors (100%) and generally high cleanliness (90.2%) (Table 2).

In total, 141 soil samples were tested for intestinal parasitic contamination. Of these, 58 (41.1%) were from schools in the Phru Nai subdistrict, 44 (31.2%) from the Koh Yao Noi subdistrict, and 39 (27.7%) from the Koh Yao Yai subdistrict (Fig 5). The most common sampling

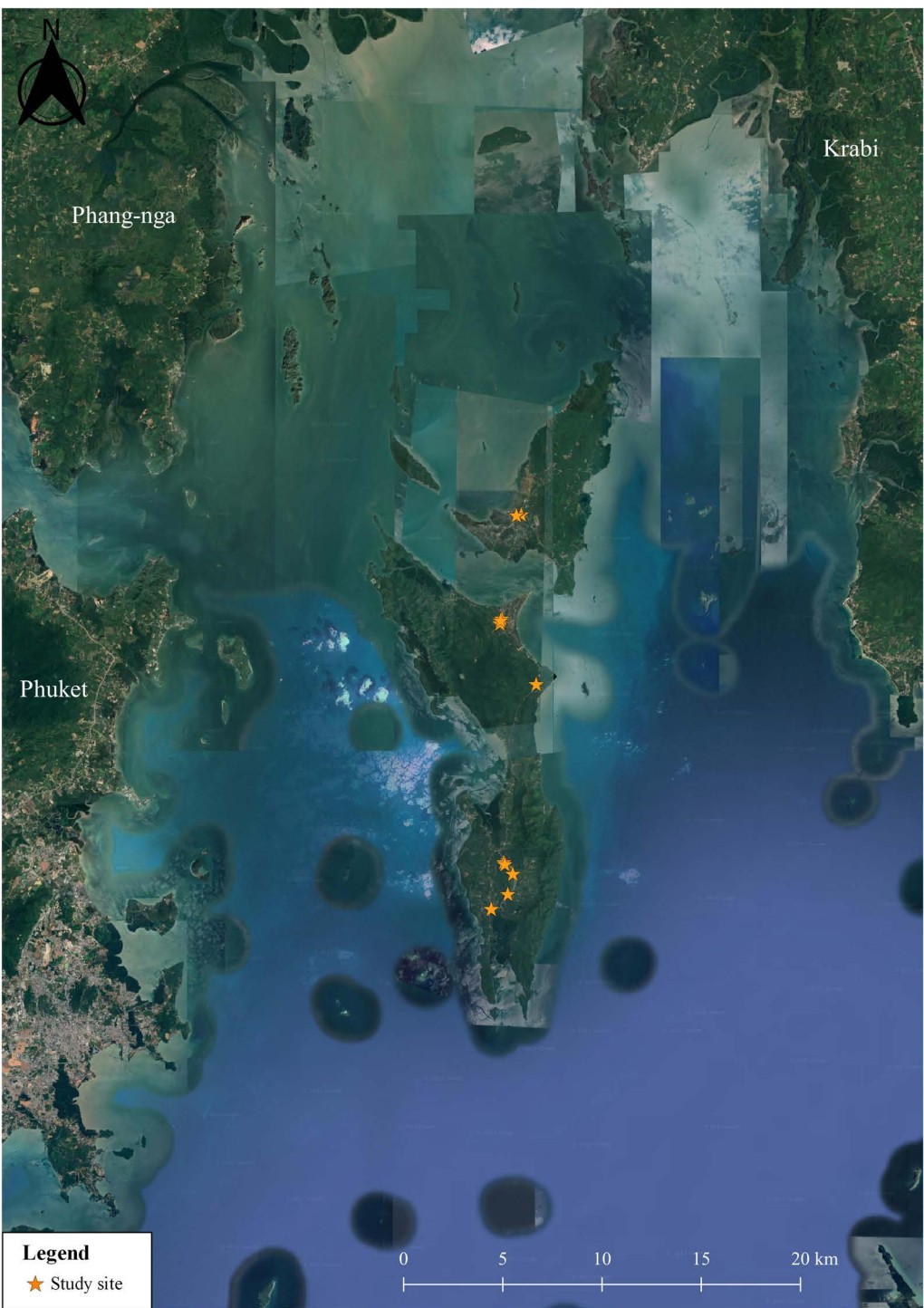

**Fig 4. Location of vegetable sample collection sites in Koh Yao District, Phang Nga Province.** Quantum GIS version 3.16 (ESRI satellite) was used to generate the map (https://qgis.org/en/site/; accessed on 31 July 2024).

**Table 2. General information on fresh vegetable samples (n = 133).**

| Characteristic | Number of samples (n = 133) | Percentage (%) |
|---|---|---|
| Subdistrict | | |
| Koh Yao Noi | 42 | 31.6 |
| Koh Yao Yai | 39 | 29.3 |
| Phru Nai | 52 | 39.1 |
| Types of fresh vegetables | | |
| Cucumber | 5 | 3.8 |
| Thai basil | 8 | 6.0 |
| Gotu kola | 4 | 3.0 |
| Bok choy | 3 | 2.3 |
| Holy basil | 6 | 4.5 |
| Cabbage | 13 | 9.8 |
| Celery | 4 | 3.0 |
| Leek | 12 | 9.0 |
| Bean sprout | 2 | 1.5 |
| Yardlong bean | 6 | 4.5 |
| Winged bean | 3 | 2.3 |
| Baby cabbage | 2 | 1.5 |
| Chinese cabbage | 10 | 7.5 |
| Lettuce | 9 | 6.8 |
| Coriander | 6 | 4.5 |
| Culantro | 9 | 6.8 |
| Chinese morning glory | 10 | 7.5 |
| Bitter gourd | 3 | 2.3 |
| Peppermint | 2 | 1.5 |
| Okra | 2 | 1.5 |
| Thai eggplant | 14 | 10.5 |
| Place | | |
| Market | 26 | 19.6 |
| Store | 107 | 80.4 |
| Source | | |
| Planted on the island | 76 | 57.1 |
| Imported | 57 | 42.9 |
| Import source (n = 76) | | |
| Krabi | 41 | 53.9 |
| Nakhon Si Thammarat | 11 | 14.5 |
| Phuket | 20 | 26.3 |
| Surat Thani | 4 | 5.3 |
| Have animals | | |
| No | 122 | 91.7 |
| Yes | 11 | 8.3 |
| Animals found (n = 11) | | |
| Cat | 11 | 100.0 |
| Area characteristics | | |
| Cement | 133 | 100.0 |
| Hygiene | | |
| No | 13 | 9.8 |

*(Continued)*

**Table 2.** (Continued)

| Characteristic | Number of samples (n = 133) | Percentage (%) |
|---|---|---|
| Yes | 120 | 90.2 |

locations were the cafeteria, playgrounds, and football fields, each accounting for 17.7% of the total. This was followed by the bathroom/prayer room areas, sidewalks, and school grounds, each accounting for 14.9%, and vegetable plots, which contributed 2.1%. All the schools had pets, with cats being the most common (92.9%), followed by chickens (19.2%), dogs (15.6%), goats (8.5%), and rats (7.1%). All soil samples were characterized as dry (100.0%), as shown in Table 3.

## Intestinal parasite contamination of fresh vegetables

The sedimentation method detected two types of parasites: *Ascaris* spp. eggs and hookworm larvae (Fig 6). Of the samples tested, three (2.3%) were contaminated with intestinal parasites: two (1.5%) with *Ascaris* spp. eggs and one (0.8%) with hookworm larvae. Among the 21 types of fresh vegetables examined, contamination was found in two samples (20.0%) of Chinese morning glory, with each sample (10.0%) showing contamination with roundworms and hookworm larvae, respectively. One peppermint sample was contaminated and had *Ascaris* spp. eggs (Table 4).

When parasite contamination in fresh vegetables was classified by area across the three subdistricts, the highest contamination rate was found in the Koh Yao Noi subdistrict, with three samples (7.1%) testing positive for parasites. No contamination was detected in samples from the Koh Yao Yai or Phru Nai subdistricts. Among the 21 types of fresh vegetables from the Koh Yao Noi subdistrict, two samples of Chinese morning glory were contaminated (40.0%), and one sample of peppermint was also contaminated (50.0%), as detailed in Table 5.

## Intestinal parasite contamination of soil samples

This study investigated intestinal parasite contamination of soil from public schools in the Koh Yao District. Contamination by one type of parasite (*Toxocara* spp.) was identified (Fig 7). Intestinal parasitic contamination was detected in 11 (84.6%) of the 13 public schools in Koh Yao District, Phang Nga Province. Out of 141 soil samples, 42 (29.8%) tested positive for *Toxocara* eggs. The highest contamination level was observed in Phru Nai subdistrict (50.0%), followed by Koh Yao Yai subdistrict (20.5%) and Koh Yao Noi subdistrict (11.4%; 95% CI: 4.5–23.1). The overall differences in the contamination rates among the subdistricts were significant ($p < 0.05$), as shown in Table 6.

Among the specimen collection sites, football fields exhibited the highest rate of *Toxocara* egg contamination at 36.0%, followed by areas around toilet/prayer rooms, vegetable plots (33.3%), sidewalks (28.6%), schoolyards (28.6%), and playgrounds (28.0%). The lowest contamination rate was found in areas around cafeterias (24.0%) as shown in Table 7. There were no significant differences in the distribution of *Toxocara* eggs among the collection sites ($p > 0.05$).

Based on the season of specimen collection, the dry season had the highest *Toxocara* egg contamination rate at 48.6%, whereas the wet season had a contamination rate of 9.0%. Significant differences in the distribution of *Toxocara* eggs were observed between seasons ($p < 0.05$) (Table 8).

## Discussion

Consumption of raw vegetables contributes significantly to the transmission of parasites to humans. Identifying parasites in vegetables allowed us to clarify the possible sources of harmful parasite acquisition in the study region. This study revealed a parasite contamination prevalence of 2.3% in vegetable samples from Koh Yao, Phang Nga Province, which

is comparable with the 5.9% prevalence reported in Ankara, Turkey [35]. However, many countries have higher contamination rates ranging from 14% to 58%. The countries include Iran [36,37], Egypt [38], Vietnam [39], Poland [25], Thailand [1], Libya [40], Brazil [41], and the Philippines [20]. The prevalence of STHs is higher in warm seasons than in cold seasons [18,42,43]. Our findings can be explained by the fact that most of the locations where vegetable samples were collected were shops with clean areas and cemented stalls, where animals were rarely present. Furthermore, the low prevalence and its variation may be attributed to the diagnostic test used, environmental factors (e.g., soil pH, soil texture), geographical location,

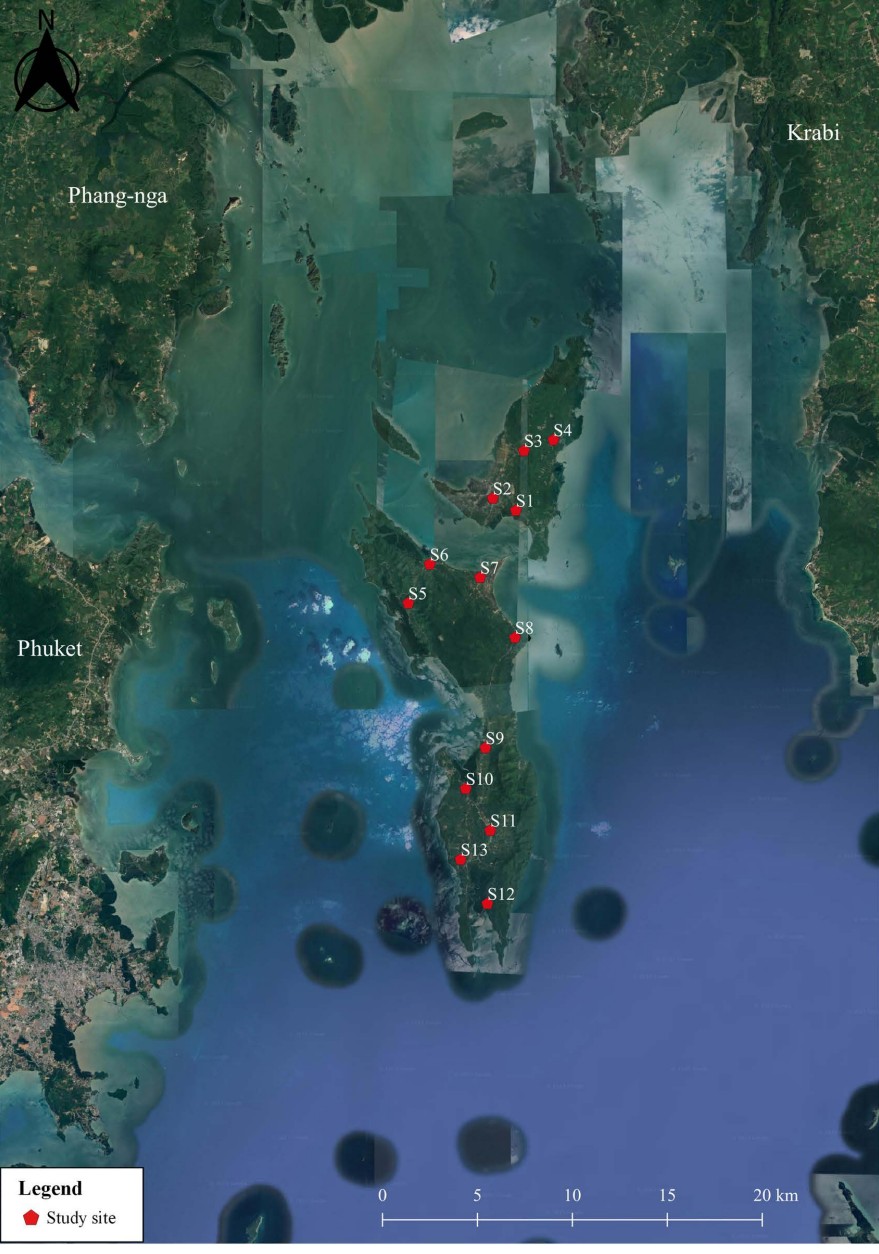

**Fig 5. Location of soil sample collection sites in Koh Yao District, Phang Nga Province.** Quantum GIS version 3.16 (ESRI satellite) was used to generate the map (https://qgis.org/en/site/; accessed on 31 July 2024).

and differences in the shape and surface of vegetables [40,44]. Vegetables with uneven surfaces facilitate the adhesion of ova, cysts, and oocysts of parasites more than those with smooth surfaces [44]. The observed differences may also result from variations in the types and quantities of vegetables evaluated.

Peppermint and Chinese morning glory were identified as contaminated samples in this study with contamination rates of 50.0% and 20.0%, respectively. These results contradict those of prior studies, which found that lettuce had the highest rate of parasitic infection [2,17,41]. In Thailand, peppermint is commonly used seasonally to garnish various dishes. The high contamination rate observed in peppermint could be due to the coarse texture of its leaves, in which contaminants can become trapped [21]. In contrast, the Chinese morning glory was found to have a lower parasite contamination, which could be due to the smooth surface of its stems, which can reduce the likelihood of parasites attaching [1].

**Table 3. General information on soil samples (n = 141).**

| Characteristic | Number of samples (n = 141) | Percentage (%) |
|---|---|---|
| Subdistrict | | |
| Koh Yao Noi | 44 | 31.2 |
| Koh Yao Yai | 39 | 27.7 |
| Phru Nai | 58 | 41.1 |
| Place | | |
| Football fields | 21 | 14.9 |
| Areas around toilet/prayer room | 25 | 17.7 |
| Vegetable plot | 3 | 2.1 |
| Sidewalks | 21 | 14.9 |
| Schoolyards | 21 | 14.9 |
| Playgrounds | 25 | 17.7 |
| Areas around cafeterias | 25 | 17.7 |
| Have animals | | |
| Yes | 141 | 100.0 |
| Cat | | |
| No | 10 | 7.1 |
| Yes | 131 | 92.9 |
| Dog | | |
| No | 119 | 84.4 |
| Yes | 22 | 15.6 |
| Chicken | | |
| No | 114 | 80.8 |
| Yes | 27 | 19.2 |
| Goat | | |
| No | 129 | 91.5 |
| Yes | 12 | 8.5 |
| Rat | | |
| No | 131 | 92.9 |
| Yes | 10 | 7.1 |
| Soil characteristic | | |
| Dry | 141 | 100.0 |

*Ascaris* spp. eggs were discovered in 1.5% (2/133) of vegetable samples; hence, they were the most common pathogenic parasite in this study. The contamination rate of *Ascaris* spp.

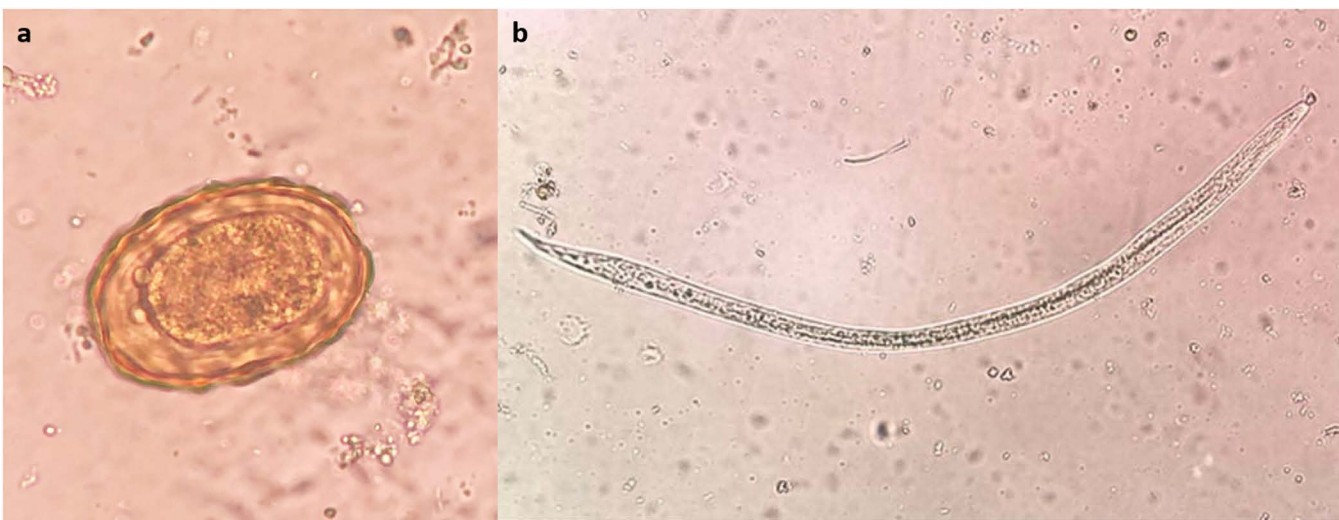

**Fig 6. Representative images of parasites found in this survey.** (a) *Ascaris* spp. eggs and (b) hookworm larvae. Original magnification: 400×.

**Table 4. Distribution of intestinal parasites in fresh vegetable samples collected from three subdistricts in Koh Yao District, Phang Nga Province, Thailand (n = 133).**

| Vegetable type | Number of samples | *Ascaris* spp. eggs (%) | Hookworm eggs (%) | Number positive for contamination (%)* |
|---|---|---|---|---|
| Cucumber | 5 | 0 | 0 | 0 |
| Thai basil | 8 | 0 | 0 | 0 |
| Gotu kola | 4 | 0 | 0 | 0 |
| Bok choy | 3 | 0 | 0 | 0 |
| Holy basil | 6 | 0 | 0 | 0 |
| Cabbage | 13 | 0 | 0 | 0 |
| Celery | 4 | 0 | 0 | 0 |
| leek | 12 | 0 | 0 | 0 |
| Bean sprout | 2 | 0 | 0 | 0 |
| Yardlong bean | 6 | 0 | 0 | 0 |
| Winged bean | 3 | 0 | 0 | 0 |
| Baby cabbage | 2 | 0 | 0 | 0 |
| Chinese cabbage | 10 | 0 | 0 | 0 |
| Lettuce | 9 | 0 | 0 | 0 |
| Coriander | 6 | 0 | 0 | 0 |
| Culantro | 9 | 0 | 0 | 0 |
| Chinese morning glory | 10 | 1 (10.0) | 1 (10.0) | 2 (20.0) |
| Bitter gourd | 3 | 0 | 0 | 0 |
| Peppermint | 2 | 1 (50.0) | 0 | 1 (50.0) |
| Okra | 2 | 0 | 0 | 0 |
| Thai eggplant | 14 | 0 | 0 | 0 |
| **Overall** | **133** | **2 (1.5)** | **1 (0.8)** | **3 (2.3)** |

*The number of positive parasitic contamination events was significantly different among types of vegetables ($p < 0.05$).

eggs have been reported to reach 68.0% in Tripoli, Libya [40]; 20.8% in Arba Minch, southern Ethiopia (16); 14.3% in Ibadan, Nigeria [44]; 12.3% in Jimma, Ethiopia [45]; 8.1% in Shahre-kord, Iran [43]; and 8.7% in the northeast [21] and 2.6% in the south [1] of Thailand. Contamination by STHs occurs at various stages including planting, harvesting, transportation, and marketing. Variations in contamination levels may be because of differences in the soil type, planting and irrigation water quality, and hygiene practices during marketing [16,17,45].

In this present study, hookworms were detected in 0.8% (1/133) of the examined vegetables. This result is consistent with other studies conducted in southern Thailand (16.3%) [1,13], Ghana (13%) [13], northeastern Thailand (6.3%) [21], Sudan (5.7%) [17] and northern Iran (4.4%) [18]. In contrast, no hookworm species were found in some previous studies [16,44,45]. The presence of hookworm contamination in this study may be linked to poor sanitation, particularly when hygienic standards are compromised at any stage, from planting to sale.

The rates of parasitic contamination in raw vegetables varied significantly among samples collected from different sub-districts. Samples from Koh Yao Noi subdistrict had the highest contamination rates, while no contamination was detected in samples from Koh Yao Yai and Phru Nai subdistricts. The differences among the three districts may be attributed to variations in vegetable sources, the number of shops and vegetable samples, and the vendors' methods of handling and washing vegetables hygienically.

**Table 5. Distribution of intestinal parasites in fresh vegetable samples among three subdistricts in Koh Yao District, Phang Nga Province, Thailand (n = 133).**

| Vegetable type | Number of examined samples | | Koh Yao Noi | | Koh Yao Yai | | Phru Nai | |
|---|---|---|---|---|---|---|---|---|
| | Examined | Positive (%) | Examined | Positive (%) | Examined | Positive (%) | Examined | Positive (%) |
| Cucumber | 5 | 0 | 2 | 0 | 1 | 0 | 2 | 0 |
| Thai basil | 8 | 0 | 2 | 0 | 4 | 0 | 2 | 0 |
| Gotu kola | 4 | 0 | 2 | 0 | 1 | 0 | 1 | 0 |
| Bok choy | 3 | 0 | 0 | 0 | 0 | 0 | 3 | 0 |
| Holy basil | 6 | 0 | 3 | 0 | 2 | 0 | 1 | 0 |
| Cabbage | 13 | 0 | 2 | 0 | 4 | 0 | 7 | 0 |
| Celery | 4 | 0 | 1 | 0 | 2 | 0 | 1 | 0 |
| leek | 12 | 0 | 2 | 0 | 4 | 0 | 6 | 0 |
| Bean sprout | 2 | 0 | 1 | 0 | 1 | 0 | 0 | 0 |
| Yardlong bean | 6 | 0 | 2 | 0 | 1 | 0 | 3 | 0 |
| Winged bean | 3 | 0 | 1 | 0 | 0 | 0 | 2 | 0 |
| Baby cabbage | 2 | 0 | 0 | 0 | 1 | 0 | 1 | 0 |
| Chinese cabbage | 10 | 0 | 3 | 0 | 4 | 0 | 3 | 0 |
| Lettuce | 9 | 0 | 4 | 0 | 3 | 0 | 2 | 0 |
| Coriander | 6 | 0 | 1 | 0 | 2 | 0 | 3 | 0 |
| Culantro | 9 | 0 | 4 | 0 | 2 | 0 | 3 | 0 |
| Chinese morning glory | 10 | 2 (20.0) | 5 | 2 (40.0) | 2 | 0 | 3 | 0 |
| Bitter gourd | 3 | 0 | 0 | 0 | 0 | 0 | 3 | 0 |
| Peppermint | 2 | 1 (50.0) | 2 | 1 (50.0) | 0 | 0 | 0 | 0 |
| Okra | 2 | 0 | 0 | 0 | 0 | 0 | 2 | 0 |
| Thai eggplant | 14 | 0 | 5 | 0 | 5 | 0 | 4 | 0 |
| **Overall** | **133** | **3 (2.3)** | **42** | **3 (7.1)\*** | **39** | **0\*** | **52** | **0\*** |

*The number of positive parasitic contamination events was significantly different among the three districts ($p < 0.05$).

Soil contaminated with *Toxocara* eggs has been reported in various tropical and subtropical regions. However, data regarding *Toxocara* egg contamination in southern Thailand are limited. This study found that 42 of 141 (29.8%) soil samples from public schools in southern Thailand were contaminated with *Toxocara* eggs. This rate is higher than that in other countries, such as 7% in Ardabil city, northwestern Iran [46], 4.75–12.84% in public places in India [47,48], 14.9% in urban and rural areas of Poland [49], 20.4% in children's play areas in southern England [50], and 28.6% in public parks in Isfahan city in central Iran [51]. However, it was lower than the rates observed in the Philippines (43%) [52,53], Ahvaz, southwestern Iran (45%) [54], Greater Lisbon, Portugal (53%) [31], parks in northeastern Poland (32–46%) [55], and the urban and suburban areas of Malaysia (45.8–54.5%) [33]. In our study, the contamination rate (29.8%) was comparable with the 21% observed in southeastern Asia and the global pooled rate of 21% reported in a systematic review [56]. Slightly lower contamination rates were reported in Thailand: 18.3% in Nakhon Si Thammarat Province [26] and 19% in Songkhla Province [57]. The discrepancies in contamination rates across studies may be a result of factors such as cultural practices, geographical parameters, climatic conditions, seasonal variations, soil types, pet populations, public attitudes toward pets, and differences in sample collection, examination methods, and diagnostic techniques [49,58–61].

This study highlights the significant differences between the distribution of *Toxocara* eggs in the subdistricts of Koh Yao. The findings showed that the Phru Nai subdistrict had the highest contamination rate, followed by Koh Yao Yai, with Koh Yao Noi showing the lowest contamination. These differences can be attributed to various factors, including urban-rural status (Koh Yao Noi is an urban area, whereas Phru Nai and Koh Yao Yai are rural areas), socioeconomic conditions, soil type, and stray dog and cat populations in each area [62,63].

This study found no significant differences between the distribution of *Toxocara* eggs in different specimen collection sites. The highest contamination rate was observed on football fields. In southeastern Asia in the Philippines, 42% of soil samples from a public school tested positive for *Toxocara* eggs and 49% of serum samples from children tested positive for *Toxocara* infection [64]. Hence, a positive correlation was found between *Toxocara* egg

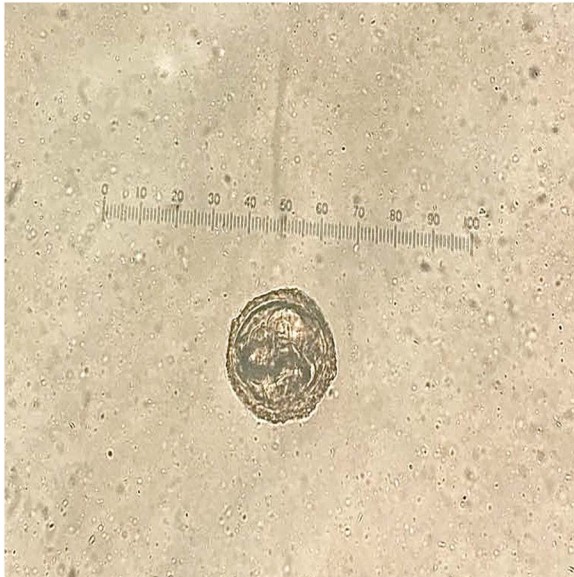

**Fig 7. Representative images of *Toxocara* spp. egg found in soil samples.** Original magnification: 400×.

**Table 6. Soil contamination by *Toxocara* eggs in the subdistricts and public schools of Koh Yao District, Phang Nga Province in southern Thailand (n = 141).**

| Study site | Number of soil samples | Number of positive samples | Rate of contamination (%) | 95% CI | p-value |
|---|---|---|---|---|---|
| Koh Yao Noi | | | | | <0.001 |
| S1 | 10 | 2 | 20.0 | | |
| S2 | 12 | 1 | 8.3 | | |
| S3 | 12 | 2 | 16.7 | | |
| S4 | 10 | 0 | 0 | | |
| Total | 44 | 5 | 11.4 | 4.5, 23.1 | |
| Koh Yao Yai | | | | | |
| S5 | 7 | 1 | 14.3 | | |
| S6 | 12 | 3 | 25.0 | | |
| S7 | 12 | 4 | 33.3 | | |
| S8 | 8 | 0 | 0 | | |
| Total | 39 | 8 | 20.5 | 10.2, 35.0 | |
| Phru Nai | | | | | |
| S9 | 10 | 5 | 50.0 | | |
| S10 | 12 | 6 | 50.0 | | |
| S11 | 12 | 6 | 50.0 | | |
| S12 | 12 | 5 | 41.7 | | |
| S13 | 12 | 7 | 58.3 | | |
| Total | 58 | 29 | 50.0 | 37.4, 62.6 | |
| **Overall** | **141** | **42** | **29.8** | **22.7, 37.7** | |

95% CI, 95% confidence interval

**Table 7. Contamination rate of intestinal parasites in the different sampling areas (n = 141).**

| Collection site | Number of soil samples | Number of positive samples | Rate of contamination (%) | 95% CI | p-value |
|---|---|---|---|---|---|
| Football fields | 25 | 9 | 36.0 | 19.5, 55.5 | 0.979 |
| Areas around toilet/prayer room | 21 | 7 | 33.3 | 16.3, 54.6 | |
| Vegetable plot | 3 | 1 | 33.3 | 3.9, 82.3 | |
| Sidewalks | 21 | 6 | 28.6 | 12.9, 49.7 | |
| Schoolyards | 21 | 6 | 28.6 | 12.9, 49.7 | |
| Playgrounds | 25 | 7 | 28.0 | 13.5, 47.3 | |
| Areas around cafeterias | 25 | 6 | 24.0 | 10.7, 42.9 | |
| **Overall** | **141** | **42** | **29.8** | **22.7, 37.7** | |

95% CI, 95% confidence interval

concentrations and the seroprevalence of *Toxocara* infection [64]. One probable reason for these findings is the growing number of stray dogs and cats in rural regions that roam freely around schools and defecate on football fields.

This study revealed significant differences between the distribution of *Toxocara* eggs between seasons. A higher incidence was observed during the dry season than during the wet season. This result aligns with other studies conducted in Nigeria [65,66] and Colombia [66], which reported an increased prevalence during the dry season. The higher prevalence during this season can be attributed to the warm temperatures that favor the survival and

**Table 8. Contamination rate of intestinal parasites in the different seasons (n = 141).**

| Season | Number of soil samples | Number of positive samples | Rate of contamination (%) | 95% CI | p-value |
|---|---|---|---|---|---|
| Dry season | 74 | 36 | 48.6 | 37.5, 59.9 | <0.001 |
| Wet season | 67 | 6 | 9.0 | 3.8, 17.5 | |
| **Overall** | **141** | **42** | **29.8** | **22.7, 37.7** | |

95% CI, 95% confidence interval

development of eggs. In our study, these factors could also explain the higher parasite prevalence and diversity observed during the dry season. The development of STH eggs is influenced by several factors, including optimal temperature ranges (20–30 °C), humidity, soil pH, depth, and texture [67,68].

The results of this study highlight that raw vegetables from local markets display low parasitic contamination rates, indicating an adequate process of planting, harvesting, transport, and handling. Nevertheless, a small proportion still carry infective helminths, which could serve as a transmission route to humans. Standard washing procedures are effective in reducing helminth contamination of raw vegetables [35,36,69]. However, in Iran, pre-washing with tap water or underground water alone did completely remove parasites [43]. Therefore, health authorities must educate the public about effective washing methods to prevent parasitic transmission. Furthermore, our findings showed that football grounds may be a major source of *Toxocara* egg contamination, particularly among young adults. We recommend that schools limit the access of dogs and cats to playgrounds and other schools. To minimize the spread of *Toxocara* infection, schools should educate students about the dangers of soil contact and discourage geophagia. Implementation of these criteria is critical for establishing effective methods to reduce the spread of toxocariasis.

This study had several limitations. First, the sample size for each vegetable varied considerably. Second, vegetables are grown using various methods; some directly contact the soil, grow within the soil, or do not contact the soil, and some are grown hydroponically. These differences were not accounted for and could serve as confounding factors. Additionally, the detection method used (microscopy) had low sensitivity and specificity. Therefore, biomolecular techniques, such as polymerase chain reaction (PCR) or real-time PCR, should be employed to detect specific species, particularly hookworms and *Strongyloides stercoralis*, which may be difficult to distinguish from naturally occurring roundworms. This study also lacked information on other factors that could influence contamination rates, such as soil composition, climatic conditions, and the absence of simultaneous analysis of the trend of intestinal parasite infections among schoolchildren or the general public in the study area. Future research should focus on conducting Knowledge, Attitude, and Practices (KAP) surveys to assess community awareness, attitudes, and behaviors regarding parasite prevention. Such surveys will help identify knowledge gaps and guide targeted interventions. Additionally, studies should explore the development of educational campaigns tailored to the local cultural and environmental context, emphasizing sustainable practices such as proper hygiene, food handling, and soil management. Addressing these issues in future research may provide a more comprehensive understanding of parasitic infections and their consequences, contributing to sustainable parasite prevention.

## Conclusion

This study demonstrated that there is a low prevalence of parasitic contamination of vegetables in the study area. Although the parasitic contamination rate was low, it might represent a transmission vector for intestinal parasites to consumers. Furthermore, this study revealed a high

prevalence of parasitic contamination of soil samples from public schools, and football fields were the most extensively affected areas. To mitigate this risk, it is imperative to educate schoolchildren and local residents on preventive measures, such as the thorough washing and cooking of vegetables before consumption, following proper handwashing procedures, and avoiding the ingestion of soil. Additionally, comprehensive health education initiatives should be offered, and hygienic practices, such as wearing gloves and washing hands after handling vegetables, should be promoted among both sellers and farmers. Measures should also be taken to prevent animals from roaming freely around school premises to reduce the spread of contamination and prevent future infections.

## Supporting information

**S1 File. Vegetable data.**
(XLSX)

**S2 File. Soil data.**
(XLSX)

## Acknowledgments

The authors extend their gratitude to the vegetable sellers who participated in this study. They also thank the Chief District Officer, Director of the Koh Yao Public Health Office, health staff, teachers, and village health volunteers in the Koh Yao Noi, Koh Yao Yai, and Phru Nai subdistricts for their invaluable assistance.

## Author contributions

**Conceptualization:** Udomsak Narkkul, Prasit Na-ek, Chuchard Punsawad.

**Data curation:** Udomsak Narkkul.

**Formal analysis:** Udomsak Narkkul, Prasit Na-ek, Chuchard Punsawad.

**Funding acquisition:** Udomsak Narkkul, Prasit Na-ek, Chuchard Punsawad.

**Investigation:** Udomsak Narkkul, Prasit Na-ek, Chuchard Punsawad.

**Methodology:** Udomsak Narkkul, Prasit Na-ek, Chuchard Punsawad.

**Project administration:** Udomsak Narkkul, Chuchard Punsawad.

**Resources:** Udomsak Narkkul, Chuchard Punsawad.

**Validation:** Udomsak Narkkul, Aulia Rahmi Pawestri, Chuchard Punsawad.

**Visualization:** Udomsak Narkkul.

**Writing – original draft:** Udomsak Narkkul, Chuchard Punsawad.

**Writing – review & editing:** Udomsak Narkkul, Prasit Na-ek, Aulia Rahmi Pawestri, Chuchard Punsawad.

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
