## [Decision Letter · Decision Letter 0]

2 Dec 2024

PONE-D-24-44096Prevalence of intestinal parasite contamination in raw vegetables and school soil samples in Koh Yao, Phang Nga Province, southern ThailandPLOS ONE

Dear Dr. Punsawad,

Thank you for submitting your manuscript to PLOS ONE. After careful consideration, we feel that it has merit but does not fully meet PLOS ONE’s publication criteria as it currently stands. Therefore, we invite you to submit a revised version of the manuscript that addresses the points raised during the review process.

We look forward to receiving your revised manuscript.

Kind regards,

Rajendra Prasad Parajuli, PhD

Academic Editor

PLOS ONE

Journal requirements: When submitting your revision, we need you to address these additional requirements. 1. Please ensure that your manuscript meets PLOS ONE's style requirements, including those for file naming. The PLOS ONE style templates can be found at https://journals.plos.org/plosone/s/file?id=wjVg/PLOSOne_formatting_sample_main_body.pdf and https://journals.plos.org/plosone/s/file?id=ba62/PLOSOne_formatting_sample_title_authors_affiliations.pdf 2. Please amend either the title on the online submission form (via Edit Submission) or the title in the manuscript so that they are identical. 3. Please include a caption for figure 2, Fig 3, Fig 4 and Fig 5.

Additional Editor Comments:

Editors’ Comment:

Version: 1

Date: Dec 1st, 2024

MS: PONE-D-24-44096

Title: Prevalence of intestinal parasite contamination in raw vegetables and school soil samples in Koh Yao, Phang Nga Province, southern Thailand.

General Comments:

In the present work, authors examined intestinal parasite contamination in raw vegetables using sedimentation and in school soils with sucrose-based flotation, focusing on contributing factors in Koh Yao, Phang Nga, Thailand.

This study presents new data on the prevalence & association between parasite contamination in raw vegetables and contextual factors which is a very important issue to explore. However, the authors should improve manuscript with comprehensive literature synthesis to grasp most of the recent/available epidemiological knowledge to illustrate the knowledge gap, establish the rationale of the study in introduction, and interpret the results correctly in discussion. Finally, concerns about clarity and logical flow should be addressed.

Despite these concerns, owing to the valuable new data and insights provided by the study, I recommend resubmission with completely rewriting the manuscript addressing ours (editors & reviewers) suggestions.

Introduction:

1. The authors could improve with comprehensive literature synthesis to grasp most of the recent/available epidemiological knowledge to illustrate the knowledge gap, to establish the rationale of the study, specially regarding associated factors.

2. None of statements in last paragraph of introduction are being supported by literature/ references which is very critical and important.

Method:

1. Selection of sites should be justified in not randomly selected.

2. Provide Ethical Approval Number and detail.

Results

1. Since authors evaluated lots of association, scenario of chance finding is quite high. So, Bonferroni correction is to consider.

2. Tables are not well structured and I suggest authors to try to put a Table together in same page.

3. Same data are in table & text, reduce text of results by half with important findings.

Discussion:

1. “This suggests that the transmission and prevalence of parasites are influenced by climate and temperature. The differences between the results of this study and others may be attributed to variations in climatic conditions, soil types, agricultural water supplies, and sanitary measures during vegetable transportation and marketing.” This interpretation is not complete and acceptable. Discuss reason behind such differences comparing characteristics of compared studies and characteristics of your study.

2. .. “These differences may be attributed to the varying sources of vegetables and the hygiene practices employed by different vendors in handling and washing the produce.…Since authors have data on characteristics of these two areas, better to relate & discuss what study data indicated instead general statement...

3. First, consider “Bonferroni Correction”, few associations will only remain significant to focus discussion. In such discussion, focus one factor in one paragraph with supporting and contrasting previous findings with comparisons on characteristics of such study to interpret discrepancy or alignment for authors interpretations.

4. Only state conclusion based on data from your study.

Despite these concerns, due to the valuable new insights provided by the comprehensive analysis, I recommend resubmission with revisions to address these issues including all concerns and comments by 2 expert reviewers.

Reviewers' comments:

Reviewer's Responses to Questions

**Comments to the Author**

1. Is the manuscript technically sound, and do the data support the conclusions?

Reviewer #1: Yes

Reviewer #2: Yes

2. Has the statistical analysis been performed appropriately and rigorously? 

Reviewer #1: Yes

Reviewer #2: Yes

3. Have the authors made all data underlying the findings in their manuscript fully available?

Reviewer #1: Yes

Reviewer #2: Yes

4. Is the manuscript presented in an intelligible fashion and written in standard English?

Reviewer #1: Yes

Reviewer #2: Yes

5. Review Comments to the Author

Reviewer #1: Dear authors,

With regard to the manuscript ID “PONE-D-24-44096”, I would like to express my general impression, comments and suggestion which will help enhance the impact of your current paper.

o General impression of the research

The study you have carried out reflects a significant contribution to the field of public health and parasitology. The methodological details, clear presentation of results, and insightful discussion provide a comprehensive correlation between the contamination of raw vegetables and soil in the public areas. Moreover, the analysis of soil sample in two different seasons also provides the general outlook regarding the chances of contamination of the edibles like the fresh vegetables, water and other food items.

o Some point-by-point comments

1. How about the trend of raising pigs in the study area, either intensively or semi-intensively or open system? As authors have mentioned the presence of Ascaris lumbricoides eggs in the examined vegetable samples. The Ascaris lumbricoides and Ascaris suum eggs (a soil transmitted helminth found in the pigs) are morphologically similar. Please describe clearly whether the source of Ascaris eggs was only from the human excreta?

2. Mention the trend of using of human excreta as the manure, which might be another supporting information for the source of contamination of vegetables.

3. Line no: 76 and 229 – authors have mentioned earthworm. It must be ‘round worm’. Please make sure the exact term.

4. Line no: 227- authors have mentioned hookworm eggs, but the image of hookworm egg is not provided. Please provide it.

5. Line no: 233 – authors have mentioned hookworm larva. Please describe in detail how did you confirm hookworm larva? Did you perform egg culture experiments? This might be the larva of other ‘strongyle nematodes’.

6. The image of Toxocara provided in the manuscript is not clear. Please provide with the clear image of it with the scale bar (if you have measured the dimension of eggs). Basically, in morphometric diagnosis of parasite eggs or larva, the scale bar is a must.

7. Control measures: Please provide strong recommendations for control measures of the soil borne parasites as described in the paper.

8. Study limitations: Provide a more in-depth discussion of potential limitations in your study. One of the limitations is lack of the simultaneous analysis of intestinal parasite infection trend among the school children or other public in the study area.

9. Future Research Directions: Suggest specific areas for future research.

……………………………….

Reviewer #2: Title: Prevalence of intestinal parasite contamination in raw vegetables and school soil samples in Koh Yao, Phang Nga Province, southern Thailand

General Comments:

In the present work, the authors investigated the prevalence of intestinal parasites in raw vegetable and school soil samples from Koh Yao, Phang Nga Province, southern Thailand.

This study provides new data on the contamination of raw vegetables and school soil samples with intestinal parasites, and the combination of raw vegetable and soil sample analysis provides a comprehensive approach to understanding potential sources of contamination and transmission. However, the authors could improve comprehensive literature synthesis to grasp most of the recent/available epidemiological knowledge to illustrate the knowledge gap, establish the rationale of the study, and interpret the results correctly. Finally, concerns about clarity and logical flow should be addressed.

Despite these concerns, owing to the valuable new data and insights provided by the study, I recommend resubmission with revisions to address these issues.

Abstract

1. The abstract is well-organized, covering the background, objectives, methodology, results, and implications concisely.

2. The repetition of "therefore" in the opening and concluding sentences could be avoided for smoother flow.

3. I do not think it is appropriate state only prevalence result in conclusion of abstract, provide conclusion about associated factors too?

Introduction:

1. This statement needs citation “The incidence of foodborne illnesses associated with the consumption of raw vegetables has increased.” Many such sentences in introductions need to be cited.

2. Synthesis of literature for trends in prevalence and associated factors is not enough to establish knowledge gap for rationale of study with logical flow.

3. The prevalence data for southern Thailand and specific parasite contamination percentages in Nakhon Si Thammarat are valuable but could be more directly tied to the study’s relevance. For instance: ” Highlights the similarity or differences between the two regions to justify the study further.”

4. ……Line 98 - 100, The objectives could be made more concise.

Method:

1. The list of vegetable types and their botanical names can be summarized in a table or grouped by category (e.g., leafy greens, root vegetables) to reduce redundancy.

2. Provide more details about how vegetable and soil samples were distributed across subdistricts or specific vendors to illustrate representativeness.

3. Elaborate on how cleanliness and hygiene were assessed for purchasing areas and their impact on contamination levels.

4. Mention if there were any quality control measures during soil collection (e.g., sterilized tools, sampling depth consistency).

5. Specify if vegetables were washed whole or cut into smaller pieces before the washing process to ensure uniformity.

6. Indicate if the solution volume (1000 mL) was adjusted for vegetable types or sample size deviations.

Results

1. Significant value in Bold would easier to follow. May be two digits after decimal is enough.

2. Tables should be placed in same page for ease of review with single space.

3. While tables are useful, adding bar charts or pie charts to visualize key findings (e.g., contamination rates by vegetable type or subdistrict) could make the data more engaging.

4. Authors may try comparison of contamination rates between imported and locally grown vegetables, which could provide actionable insights.

Discussion:

1. The authors compared the observed prevalence in this study with the reported prevalence in the nearby region of Thailand followed by neighboring countries and the global literature to judge the prevalence and judged “low” prevalence in the study population. The author could discuss reasons such as specific agricultural practices or the impact of environmental conditions (e.g., soil pH, texture) on contamination rates to strengthen the narrative.

2. Discussion regarding prevalence and associated factors missed the logical flow with supporting & contrasting literature with authors interpretations.

3. In line 297-298, The statement implies that the smooth surface of Chinese morning glory reduces parasitic attachment, yet it is cited as a reason for the presence of parasites. These two ideas are contradictory - if the surface reduces attachment, it should theoretically result in lower contamination, not higher.

Despite these concerns, owing to the valuable new insights provided by the comprehensive analysis, I recommend resubmission with revisions to address these issues.

6. PLOS authors have the option to publish the peer review history of their article (what does this mean? ). If published, this will include your full peer review and any attached files.

**Do you want your identity to be public for this peer review?** For information about this choice, including consent withdrawal, please see our Privacy Policy .

Reviewer #1: **Yes: ** Pitambar Dhakal

Reviewer #2: **Yes: ** Zainuddin Ansari

---

## [Author Response · Author response to Decision Letter 1]

15 Jan 2025

Response: We appreciate the reviewer’s helpful comment. We have thoroughly reviewed and revised our manuscript to ensure it complies with PLOS ONE's style requirements.

2. Please amend either the title on the online submission form (via Edit Submission) or the title in the manuscript so that they are identical.

Response: Thank you for pointing this out. We have amended the title to ensure consistency between the online submission form and the manuscript.

3. Please include a caption for figure 2, Fig 3, Fig 4 and Fig 5.

Response: Thank you for your valuable feedback. We have now included captions for Figures 2, 3, 4, 5, 6 and 7 in the manuscript. Each caption provides a clear and concise description of the figures to enhance the clarity of the presentation. We appreciate your attention to this detail.

–Line 131-134, 150-159, 172-173, 220, 251, 279-280, 305-306.

Editors’ Comment:

General Comments:

In the present work, authors examined intestinal parasite contamination in raw vegetables using sedimentation and in school soils with sucrose-based flotation, focusing on contributing factors in Koh Yao, Phang Nga, Thailand.

This study presents new data on the prevalence & association between parasite contamination in raw vegetables and contextual factors which is a very important issue to explore. However, the authors should improve manuscript with comprehensive literature synthesis to grasp most of the recent/available epidemiological knowledge to illustrate the knowledge gap, establish the rationale of the study in introduction, and interpret the results correctly in discussion. Finally, concerns about clarity and logical flow should be addressed.

Despite these concerns, owing to the valuable new data and insights provided by the study, I recommend resubmission with completely rewriting the manuscript addressing ours (editors & reviewers) suggestions.

Introduction:

1. The authors could improve with comprehensive literature synthesis to grasp most of the recent/available epidemiological knowledge to illustrate the knowledge gap, to establish the rationale of the study, specially regarding associated factors.

Response: We greatly appreciate the reviewer’s suggestion. We have specifically addressed the most recent and available epidemiological knowledge to illustrate the existing knowledge gap and establish the rationale for the study. –Line 51-59.

2. None of statements in last paragraph of introduction are being supported by literature/ references which is very critical and important.

Response: We sincerely appreciate the reviewer’s feedback. We have revised the last paragraph to include relevant references that substantiate the claims made. We have incorporated recent studies that address the key points discussed in the paragraph. –Line 102-110.

Method:

1. Selection of sites should be justified in not randomly selected.

Response: We appreciate the reviewer’s comment. We have provided a detailed rationale for the selection of sites in this study, as outlined in the revised manuscript:

“Koh Yao is a remote area where transportation is primarily by boat, and access to healthcare is limited. The majority of the population is engaged in agriculture and tourism. Previous studies have reported a prevalence of intestinal parasites in humans in this region, ranging from 9.3% to 18.42% (12, 29, 30), which may be linked to the consumption of unwashed vegetables. Fresh vegetables were purchased from all markets located in the three subdistricts. These vegetables were sourced from various farms and agricultural areas, both on the island and in nearby provinces. Soil samples were collected from all schools across the three subdistricts.” –Line 123-130.

2. Provide Ethical Approval Number and detail.

Response: Thank you for your comment. We have now included the ethical approval number and details in the revised manuscript. “This study was reviewed and approved by the Human Research Ethics Committee of Walailak University, Thailand (Approval Number: WUEC-22-329-01), prior to sample collection. Informed consent was not required for the collection of raw vegetable and school soil samples as they were obtained from publicly accessible locations. Permission was obtained from the school authorities for soil sampling on the school premises and from local vendors before collecting vegetable samples.” –Line 202-207.

Results

1. Since authors evaluated lots of association, scenario of chance finding is quite high. So, Bonferroni correction is to consider.

Response: We thank the reviewer for raising this important point. In this study, we compared the rates of intestinal parasitic contamination across different types of vegetables, subdistricts, and sample collection sites. As the data are categorical data, the chi-square test was employed as the appropriate statistical method for comparing categorical variables. We acknowledge that the Bonferroni correction is primarily used to adjust for multiple comparisons in the context of continuous data, where means between groups are compared. However, in this study, our focus was on categorical data, and therefore, the Bonferroni correction was not applicable to our analysis. - Statistical analysis part.

2. Tables are not well structured and I suggest authors to try to put a Table together in same page.

Response: Thank you for your suggestion. We have reorganized the table 1-7 to fit on a single page for easier reading, ensuring that all the relevant data is clearly presented and easily accessible. –Line 148, 241, 270, 281, 291, 307, 319, 321.

3. Same data are in table & text, reduce text of results by half with important findings.

Response: Thank you for your valuable feedback. We have revised the results section by condensing the text while retaining the key findings. –- Results part.

Discussion:

1. “This suggests that the transmission and prevalence of parasites are influenced by climate and temperature. The differences between the results of this study and others may be attributed to variations in climatic conditions, soil types, agricultural water supplies, and sanitary measures during vegetable transportation and marketing.” This interpretation is not complete and acceptable. Discuss reason behind such differences comparing characteristics of compared studies and characteristics of your study.

Response: Thank you for your valuable feedback. We have revised the interpretation to provide a more comprehensive discussion of the differences between our findings and those of other studies. –Line 331-338.

2. .. “These differences may be attributed to the varying sources of vegetables and the hygiene practices employed by different vendors in handling and washing the produce.…Since authors have data on characteristics of these two areas, better to relate & discuss what study data indicated instead general statement...

Response: Thank you for your suggestion. We have revised the discussion to incorporate specific data from the study. The differences between the two areas can be linked to the characteristics identified in our data. –- Discussion part.

3. First, consider “Bonferroni Correction”, few associations will only remain significant to focus discussion. In such discussion, focus one factor in one paragraph with supporting and contrasting previous findings with comparisons on characteristics of such study to interpret discrepancy or alignment for authors interpretations.

Response: We thank the reviewer for raising this important point. In this study, we compared the rates of intestinal parasitic contamination across different types of vegetables, subdistricts, and sample collection sites. As the data are categorical data, the chi-square test was employed as the appropriate statistical method for comparing categorical variables. We acknowledge that the Bonferroni correction is primarily used to adjust for multiple comparisons in the context of continuous data, where means between groups are compared. However, in this study, our focus was on categorical data, and therefore, the Bonferroni correction was not applicable to our analysis. To further clarify our findings, we structured the discussion by dedicating a paragraph to each significant factor. For each factor, we included comparisons with previous studies, highlighting both supporting and contrasting findings. –- Discussion part.

4. Only state conclusion based on data from your study.

Response: We have revised the manuscript to ensure that conclusions are drawn solely from the data obtained in this study. –Line 436-446.

Reviewers' comments:

Reviewer #1: Dear authors,

With regard to the manuscript ID “PONE-D-24-44096”, I would like to express my general impression, comments and suggestion which will help enhance the impact of your current paper.

General impression of the research

The study you have carried out reflects a significant contribution to the field of public health and parasitology. The methodological details, clear presentation of results, and insightful discussion provide a comprehensive correlation between the contamination of raw vegetables and soil in the public areas. Moreover, the analysis of soil sample in two different seasons also provides the general outlook regarding the chances of contamination of the edibles like the fresh vegetables, water and other food items.

Some point-by-point comments

1. How about the trend of raising pigs in the study area, either intensively or semi-intensively or open system? As authors have mentioned the presence of Ascaris lumbricoides eggs in the examined vegetable samples. The Ascaris lumbricoides and Ascaris suum eggs (a soil transmitted helminth found in the pigs) are morphologically similar. Please describe clearly whether the source of Ascaris eggs was only from the human excreta?

Response: We would like to thank the reviewers for raising this important issue. In Koh Yao, almost all the population is Muslim, and no pig farming was found in the area. Additionally, this study employed a conventional method, which could not differentiate between species of Ascaris spp. Therefore, we propose changing the term "Ascaris lumbricoides eggs" to "Ascaris spp. eggs" in this study. We will also include the limitations of conventional methodology in our discussion of this research. In the future, biomolecular techniques, such as polymerase chain reaction (PCR) or real-time PCR, should be employed to detect specific species. –Line 421-424.

2. Mention the trend of using of human excreta as the manure, which might be another supporting information for the source of contamination of vegetables.

Response: We would like to thank the reviewer for suggesting the inclusion of human excreta as a potential source of contamination. However, this study did not investigate the use of human excreta as manure. Additionally, the vegetable samples in this study were collected from shops, which are not directly involved in the cultivation of the vegetables. Therefore, we did not consider human excreta as a potential source of contamination in this context.

3. Line no: 76 and 229 – authors have mentioned earthworm. It must be ‘round worm’. Please make sure the exact term.

Response: We appreciate the reviewer’s helpful comment. We have corrected the term "earthworm" to "roundworm" in lines 81 and 277 as per your suggestion.

4. Line no: 227- authors have mentioned hookworm eggs, but the image of hookworm egg is not provided. Please provide it.

Response: We would like to thank the reviewer for pointing out this issue. In this study, we only identified hookworm larvae, not hookworm eggs. We apologize for the oversight and have corrected the text to reflect this, changing "hookworm eggs" to "hookworm larvae" in the manuscript.

5. Line no: 233 – authors have mentioned hookworm larva. Please describe in detail how did you confirm hookworm larva? Did you perform egg culture experiments? This might be the larva of other ‘strongyle nematodes’.

Response: Thank you for your valuable feedback regarding the identification of hookworm larvae. In response to your query, the parasites were morphologically identified according to the Centers for Disease Control and Prevention diagnostic reference and were examined by two trained senior medical laboratory technologists. As mentioned in the Methods section of the manuscript: “After removing the supernatant, the sediment was examined under a light microscope using 100× and 400× magnification objectives to detect the eggs and larvae of parasites. The parasites were morphologically identified according to the Centers for Disease Control and Prevention diagnostic reference (32). To confirm the presence of parasites, each vegetable sample was examined by two trained senior medical laboratory technologists. Three slides per sample were prepared by two independent investigators to enhance parasite detection (1).” –Line 178-183.

6. The image of Toxocara provided in the manuscript is not clear. Please provide with the clear image of it with the scale bar (if you have measured the dimension of eggs). Basically, in morphometric diagnosis of parasite eggs or larva, the scale bar is a must.

Response: Thank you for your feedback. The image of Toxocara egg with the scale bar has inserted. Please see Fig 7.

7. Control measures: Please provide strong recommendations for control measures of the soil borne parasites as described in the paper.

Response: We thank the reviewer for the valuable comment. We have provided recommendations for control measures of soil-borne parasites as described in the conclusion. “This study demonstrated that there is a low prevalence of parasitic contamination of vegetables in the study area. Although the parasitic contamination rate was low, it might represent a transmission vector for intestinal parasites to consumers. Furthermore, this study revealed a high prevalence of parasitic contamination of soil samples from public schools, and football fields were the most extensively affected areas. To mitigate this risk, it is imperative to educate schoolchildren and local residents on preventive measures, such as the thorough washing and cooking of vegetables before consumption, following proper handwashing procedures, and avoiding the ingestion of soil. Additionally, comprehensive health education initiatives should be offered, and hygienic practices, such as wearing gloves and washing hands after handling vegetables, should be promoted among both sellers and farmers. Measures should also be taken to prevent animals from roaming freely around school premises to reduce the spread of contamination and prevent future infections.” –Line 436-446.

8. Study limitations: Provide a more in-depth discussion of potential limitations in your study. One of the limitations is lack of the simultaneous analysis of intestinal parasite infection trend among the school children or other public in the study area.

Response: We thank the reviewer for the valuable suggestion. We have now included a more in-depth discussion of the study’s limitations in the revised manuscript. Specifically, we have acknowledged the limitation regarding the lack of simultaneous analysis of the trend of intestinal parasite infections among schoolchildren and the general public in the study area. constraints.

–Line 426-427.

9. Future Research Directions: Suggest specific areas for future research.

Response: Thank you for your helpful feedback regarding the "Future Research Directions" section. In response, we have added the following suggestion: "Future research should focus on conducting Knowledge, Attitude, and Practices (KAP) surveys to assess community awareness, attitudes, and behaviors regarding parasite prevention. Such surveys will help identify knowledge gaps and guide targeted interventions. Additionally, studies should explore t

---

## [Decision Letter · Decision Letter 1]

20 Feb 2025

PONE-D-24-44096R1Prevalence of intestinal parasite contamination in raw vegetables and school soil samples in Koh Yao, Phang Nga Province, southern ThailandPLOS ONE

Dear Dr. Punsawad,

Thank you for submitting your manuscript to PLOS ONE. After careful consideration, we feel that it has merit but does not fully meet PLOS ONE’s publication criteria as it currently stands. Therefore, we invite you to submit a revised version of the manuscript that addresses the points raised during the review process.

We look forward to receiving your revised manuscript.

Kind regards,

Rajendra Prasad Parajuli, PhD

Academic Editor

PLOS ONE

**Journal Requirements:**

**Additional Editor Comments:**

Editors’ Comment:

Version: 2 Date: Feb 10, 2025

MS: PONE-D-24-44096R1

Title: Prevalence of intestinal parasite contamination in raw vegetables and school soil samples in Koh Yao, Phang Nga Province, southern Thailand.

General Comments:

We appreciate the authors' efforts in revising the manuscript based on the reviewers' and editors' comments. The revision is largely satisfactory, with only a few minor omissions that need to be addressed. As these revisions do not require further review, the manuscript is conditionally accepted and will be accepted upon completion of the required corrections. Please submit the final revised version at your earliest convenience.

Suggested Minor edits

1. 'spp.' mentioned after parasite name should not be italic, please see it once more and correct throughout manuscripts

2. Start Introduction in Page 3

3. Guide where to place which figure like Fig 1: Here; so on]

4. For better look use font size 10 for Tables and figures with single space [not 1.5 or 2]

5. Page 19, Line 378, there is comma after period, its typo, but rectify such typo throughout manuscript

6. Move reference to next page

7. Read whole manuscript one more time to rectify any typological error

Reviewers' comments:

Reviewer's Responses to Questions

**Comments to the Author**

1. If the authors have adequately addressed your comments raised in a previous round of review and you feel that this manuscript is now acceptable for publication, you may indicate that here to bypass the “Comments to the Author” section, enter your conflict of interest statement in the “Confidential to Editor” section, and submit your "Accept" recommendation.

Reviewer #1: All comments have been addressed

Reviewer #2: All comments have been addressed

2. Is the manuscript technically sound, and do the data support the conclusions?

Reviewer #1: Yes

Reviewer #2: Yes

3. Has the statistical analysis been performed appropriately and rigorously? 

Reviewer #1: Yes

Reviewer #2: Yes

4. Have the authors made all data underlying the findings in their manuscript fully available?

Reviewer #1: Yes

Reviewer #2: Yes

5. Is the manuscript presented in an intelligible fashion and written in standard English?

Reviewer #1: Yes

Reviewer #2: Yes

6. Review Comments to the Author

**Reviewer #1:**  Dear Autor

Regarding your submission (PONE-D-24-44096R1), you have satisfactorily and completely answered the queries and addressed them respectively. I would like to suggest one thing the 'spp.' mentioned after parasite name should not be italic, please see it once more and correct.

Regards,

**Reviewer #2: ** Reviewer's Report Date: 2/8/2025

Reviewer no. 1 Version: 2

Manuscript ID: PONE-D-24-44096R1

Title: Prevalence of Intestinal Parasite Contamination in Raw Vegetables and School Soil Samples in Koh Yao, Phang Nga Province, Southern Thailand

General Comments:

The revised manuscript has incorporated most of the suggested improvements, particularly in enhancing the literature synthesis, restructuring the discussion, and improving the clarity of statistical reporting. The authors have strengthened the literature review, improved methodological clarity, and refined the discussion for better logical flow. The study provides valuable new data on the prevalence of intestinal parasites in raw vegetables and school soil, which has important public health implications. The methodology is sound, the results are clearly presented, and the discussion effectively interprets the findings in a relevant epidemiological context. Given these improvements, I recommend the manuscript for acceptance.

Abstract:

1. The revised abstract is well-structured and now includes associated factors along with prevalence results, addressing previous concerns.

2. The statistical significance of contamination differences is reported correctly.

3. The conclusion appropriately emphasizes the importance of hygiene practices and preventive measures.

Introduction:

1. The introduction now presents a strong and well-referenced literature review that establishes the knowledge gap and study rationale effectively.

2. The epidemiological context of soil-transmitted helminths and zoonotic parasites is well-explained, and the justification for selecting Koh Yao as the study site is clear.

3. The objectives are concise and align well with the study’s scope.

Methods:

1. The methodology is now well-detailed, ensuring reproducibility and transparency.

2. The authors have effectively clarified the sampling strategy, data collection methods, and statistical analyses used.

3. Hygiene assessment and ethical considerations are appropriately described.

4. The revisions ensure that the approach is robust and scientifically sound.

Results:

1. The results section is now well-organized and presents key findings with clarity.

2. The authors have successfully streamlined text to reduce redundancy while ensuring that all relevant data are effectively reported.

3. The statistical significance of contamination differences is well-presented, and tables are neatly formatted.

Discussion:

1. The discussion is more structured, with clearer explanations of contamination patterns.

2. The study’s findings are contextualized within public health frameworks, and the conclusions are directly supported by the data.

3. The contradiction regarding Chinese morning glory’s smooth surface and its contamination rate has been corrected.

4. The role of climate and environmental conditions in contamination variations is now discussed.

Conclusion:

1. The conclusion is now directly based on study findings, avoiding broad generalizations.

2. The emphasis on preventive strategies and the need for hygiene education is appropriate and actionable.

3. The recommendations for future research are well-stated and relevant.

Comments for authors

I commend the authors on their responsibility for reviewers' comments and overall, they adequately addressed my concerns.

Hence, I recommend acceptance of the manuscript.

7. PLOS authors have the option to publish the peer review history of their article (what does this mean? ). If published, this will include your full peer review and any attached files.

**Do you want your identity to be public for this peer review?** For information about this choice, including consent withdrawal, please see our Privacy Policy .

Reviewer #1: **Yes: ** Pitambar Dhakal

Reviewer #2: **Yes: ** Zainuddin Ansari

---

## [Author Response · Author response to Decision Letter 2]

25 Feb 2025

POINT-BY-POINT RESPONSES TO THE REVIEWERS’ COMMENTS

General Comments:

We appreciate the authors' efforts in revising the manuscript based on the reviewers' and editors' comments. The revision is largely satisfactory, with only a few minor omissions that need to be addressed. As these revisions do not require further review, the manuscript is conditionally accepted and will be accepted upon completion of the required corrections. Please submit the final revised version at your earliest convenience.

Suggested Minor edits:

1. 'spp.' mentioned after parasite name should not be italic, please see it once more and correct throughout manuscripts

Response: Thank you for your valuable feedback. We have carefully reviewed the manuscript and corrected all instances where 'spp.' was italicized after parasite names. Now, the genus and species names remain italicized, while 'spp.' is presented in regular font, as per standard scientific nomenclature.

2. Start Introduction in Page 3

Response: Thank you for your suggestion. We have adjusted the formatting of the manuscript to ensure that the Introduction now begins on Page 3 as requested. - Line 44.

3. Guide where to place which figure like Fig 1: Here; so on]

Response: Thank you for your suggestion. We have now inserted clear placement indicators within the manuscript, such as “[Fig. 1: Here],” at appropriate locations to guide the placement of each figure. - Line 133, 153, 176, 225, 260, 275, 300.

4. For better look use font size 10 for Tables and figures with single space [not 1.5 or 2]

Response: Thank you for your valuable suggestion. We have reformatted all tables to use font size 10 and all figures to use font size 16, as font size 10 was too small for the figures. Additionally, we have applied single spacing to enhance readability and maintain consistency throughout the manuscript. - Line 151, 250, 264, 278, 287, 303, 315, 317.

5. Page 19, Line 378, there is comma after period, its typo, but rectify such typo throughout manuscript

Response: Thank you for pointing this out. We have carefully reviewed the manuscript and corrected the typo on Page 19, Line 378. Additionally, we have conducted a thorough proofreading to ensure that similar typographical errors are rectified throughout the manuscript.

6. Move reference to next page

Response: Thank you for your suggestion. We have adjusted the formatting to ensure that the references are now moved to the next page as requested. - Line 473.

7. Read whole manuscript one more time to rectify any typological error

Response: Thank you for your helpful suggestion. We have carefully reread the entire manuscript and made necessary corrections to any typographical errors found. We are confident that the manuscript is now free of such errors.

Reviewer #1: Dear Autor

Regarding your submission (PONE-D-24-44096R1), you have satisfactorily and completely answered the queries and addressed them respectively. I would like to suggest one thing the 'spp.' mentioned after parasite name should not be italic, please see it once more and correct.

Response: Thank you for your positive feedback and for pointing this out. We have reviewed the manuscript once more and corrected the instances where 'spp.' was italicized after parasite names. Now, the genus and species names remain italicized, while 'spp.' is presented in regular font as per standard scientific nomenclature.

---

## [Editor Report · Decision Letter 2]

27 Feb 2025

Prevalence of intestinal parasite contamination in raw vegetables and school soil samples in Koh Yao, Phang Nga Province, southern Thailand

PONE-D-24-44096R2

Dear Dr. Punsawad,

We’re pleased to inform you that your manuscript has been judged scientifically suitable for publication and will be formally accepted for publication once it meets all outstanding technical requirements.

Kind regards,

Rajendra Prasad Parajuli, PhD

Academic Editor

PLOS ONE
---

## [Editor Report · Acceptance letter]

PONE-D-24-44096R2

PLOS ONE

Dear Dr. Punsawad,

I'm pleased to inform you that your manuscript has been deemed suitable for publication in PLOS ONE. Congratulations! Your manuscript is now being handed over to our production team.

Kind regards,

on behalf of

Dr. Rajendra Prasad Parajuli

Academic Editor

PLOS ONE